# Up-Regulation of Astrocytic Fgfr4 Expression in Adult Mice after Spinal Cord Injury

**DOI:** 10.3390/cells12040528

**Published:** 2023-02-06

**Authors:** Claire Mathilde Bringuier, Harun Najib Noristani, Jean-Christophe Perez, Maida Cardoso, Christophe Goze-Bac, Yannick Nicolas Gerber, Florence Evelyne Perrin

**Affiliations:** 1MMDN, Univ. Montpellier, EPHE, INSERM, 34095 Montpellier, France; 2UMR 5221, Univ. Montpellier, CNRS, 34095 Montpellier, France; 3Institut Universitaire de France (IUF), 75005 Paris, France

**Keywords:** spinal cord injury, astrocytes, Fgfr4, transdifferentiation, gene therapy

## Abstract

Spinal cord injury (SCI) leads to persistent neurological deficits without available curative treatment. After SCI astrocytes within the lesion vicinity become reactive, these undergo major morphological, and molecular transformations. Previously, we reported that following SCI, over 10% of resident astrocytes surrounding the lesion spontaneously transdifferentiate towards a neuronal phenotype. Moreover, this conversion is associated with an increased expression of fibroblast growth factor receptor 4 (Fgfr4), a neural stem cell marker, in astrocytes. Here, we evaluate the therapeutic potential of gene therapy upon Fgfr4 over-expression in mature astrocytes following SCI in adult mice. We found that Fgfr4 over-expression in astrocytes immediately after SCI improves motor function recovery; however, it may display sexual dimorphism. Improved functional recovery is associated with a decrease in spinal cord lesion volume and reduced glial reactivity. Cell-specific transcriptomic profiling revealed concomitant downregulation of Notch signaling, and up-regulation of neurogenic pathways in converting astrocytes. Our findings suggest that gene therapy targeting Fgfr4 over-expression in astrocytes after injury is a feasible therapeutic approach to improve recovery following traumatism of the spinal cord. Moreover, we stress that a sex-dependent response to astrocytic modulation should be considered for the development of effective translational strategies in other neurological disorders.

## 1. Introduction

Traumatic spinal cord injuries (SCI) account for over 750,000 new cases annually worldwide [1]. Depending on the level and severity of the lesion, SCI induce minor-to-severe impairments of motor, sensory, and vegetative functions. Drastic changes in physiological homeostasis occur immediately after SCI and a robust inflammatory response, driven by resident microglia and infiltrating macrophages, takes place within minutes following lesion (for review see [2]). The glial scar, mainly composed of microglia and astrocytes, has both beneficial and detrimental effects. Although reactive astrogliosis was considered detrimental to axonal regeneration, recent studies suggest that astrogliosis may also exert beneficial functions (for reviews, see [3,4,5,6]).

Cell-specific transcriptomic analysis uncovered a diverse astrocyte contribution during the central nervous system’s (CNS) development [7], synaptogenesis [8], aging [9], Alzheimer’s disease [10], ischemic stroke [11], and SCI [12]. We have previously characterized the molecular response of astrocytes at multiple time points after different lesion severity of the spinal cord [12,13]. We uncovered SCI-induced endogenous astrocyte-to-neuron conversion where over 10% of resident reactive astrocytes express classical neuronal progenitor proteins including βIII-tubulin and doublecortin with typical transformation into immature neuronal morphology [13]. Importantly, we identified the neural stem cell marker fibroblast growth factor receptor 4 (Fgfr4) as a potential endogenous modulator of astrocyte-to-neuron conversion following SCI. Astrocyte-to-neuron conversion was associated with increased Fgfr4 mRNA in reactive astrocytes and over-expression of FGFR4 protein in the cytoskeleton of elongated astrocytes situated within the lesion vicinity over the course of 6 weeks after injury. Fibroblast growth factors receptors (FGFR 1–4) are essential during progenitor cell proliferation and neuronal differentiation (for review, see [14]). FGF4, an endogenous ligand of FGFR4 promotes the differentiation of embryonic stem cells towards neural lineage [15] and is required, though not sufficient, for astrocyte dedifferentiation into neural stem cells [16]. In zebrafish, both FGFR3 and FGFR4 are required for expansion of astrocytic processes [17]. In the presence of epidermal growth factor (EGF) and basic fibroblast growth factor (bFGF), mature astrocytes from diverse CNS regions including the spinal cord can form neurospheres and further differentiate into neurons in vitro [18]. Promoting astrocyte differentiation to neurons in vivo is, thus, a promising therapeutic approach to generate new neurons in situ and counteract neuronal demise due to neurological diseases. In vivo viral delivery of the three neural conversion factors achaete-scute complex-like 1 (Ascl1), brain-2 (Brn2a), and myelin transcription factor-like 1 (Myt1l) [19] or a cocktail of Ascl1, Lmx1a, and Nurr1 [20,21] in the striatum triggers astrocyte conversion into neurons. In fact, Ascl1 alone is sufficient for in vivo astrocyte-to-neuron conversion within the dorsal midbrain, striatum, and somatosensory cortex [22]. Enforced astrocytic expression of the transcription factor Sox2 can also transform mature astrocytes into proliferative neuroblasts, and subsequently functional cortical neurons [23,24]. In addition, adeno-associated virus type 9 (AAV9)-mediated astrocytic over-expression of NeuroD1 promotes astrocyte-to-neuron conversion in the striatum [25]. Moreover, astrocyte-to-neuron conversion induced by NeuroD1 rescues neuronal loss in the temporal lobe and ameliorates neurological deficits associated with epilepsy [26]. Combination of microRNA miR218 with Neurod1, Ascl1, and Lmx1a led to the in vivo conversion of astrocytes into dopaminergic neurons in a mouse model of Parkinson’s disease [27]. In the adult injured mouse spinal cord, enforced astrocytic Sox2 expression promotes their conversion into doublecortin-positive cells [28] through a p53-dependent pathway [29], although its effect on lesion evolution, glial cells reactivity, and functional recovery remains unknown. Recently, AAV NeuroD1-base gene therapy converted reactive astrocytes into functional neurons following SCI, although its effects on motor recovery were not assessed [30]. Moreover, enforced expression of the zinc-finger transcription factor (Zfp521), predominantly in astrocytes after SCI in rats, led to the conversion of Zfp521-expressing cells into βIII-tubulin positive cells in rats [31]. Importantly, Zfp521-expression was associated with improved motor recovery and functional integrity of the spinal cord [31].

To date, astrocyte-to-neuron conversion has been achieved through the enforced expression of ectopic genes or fate-determining transcription factors that are not typically expressed by astrocytes. Here, we took an alternative approach by targeting Fgfr4 that is endogenously over-expressed in reactive astrocytes after SCI [13]. We used a lentiviral-approach to amplify the SCI-induced Fgfr4 over-expression in astrocytes. Fgfr4 over-expression in reactive astrocytes immediately after SCI in adult mice led to an improvement in motor function recovery. This was associated with condensed spinal cord lesion, myelin preservation, and reduced glial reactivity without affecting the muscular component. Interestingly, we also uncovered a possible dimorphic response with a differential functional recovery in female and in male mice. Finally, cell-specific transcriptomic analysis of SCI-induced βIII-tubulin-expressing astrocytes revealed simultaneous down-regulation of Notch signaling and up-regulation of neurogenic pathways, confirming their active engagement toward neuronal conversion.

## 2. Materials and Methods

### 2.1. Animals

Mice were housed in hygrometry and a temperature-controlled environment with a 12 h light/dark cycle. Three-month old C57BL6/6J male and female mice (Charles River, Wilmington, NC, USA) were used for behavioral studies, ex vivo diffusion weighted MRI (DW-MRI), and immunohistochemistry. For transcriptomic experiments, three-month old males transgenic mice expressing eGFP (enhanced green fluorescent protein) in astrocytes (Aldh1l1-EGFP)OFC789Gsat/Mmucd) were obtained from the Mutant Mouse Regional Resource Centre (MMRRC) and maintained on a Swiss Webster strain background. Aldehyde dehydrogenase 1 family member L1 is a specific pan-astrocytic marker [7].

### 2.2. Viral Constructs

Replication-deficient self-inactivating pHIV lentiviral vectors were obtained from the vectorology platform of Montpellier (PVM, Institute for Functional Genomics, Montpellier, France). Embryonic human kidney (HEK-293) cells were co-transduced with the vector plasmid and plasmid encoding the vesicular stomatitis virus envelope glycoprotein (VSV-G). HIV p24 gag protein was quantified for each stock via ELISA at 99 µg/mL (pHIV pGFAP Fgfr4 mCherry vector) and 92 µg/mL (pHIV pGFAP mCherry vector) (Figure 1A). The experimental group received the pHIV pGFAP Fgfr4 mCherry vector, where the viral plasmid expresses Fgfr4 and mCherry sequences under the human GFAP promotor. The control group received a pHIV pGFAP mCherry vector, where the viral plasmid expresses only the mCherry sequence.

### 2.3. Spinal Cord Injury and Viral Vector Injections

Animals were anesthetized via inhalation of isoflurane (Vetflurane**^®^**, Virbac, France, induction 4% and maintenance 2%) with 1 L/min oxygen flow rate. Eye gel was applied to the cornea to avoid drying during the surgery. The surgical area was shaved and cleaned with 70% ethanol, and the skin and muscles were incised. Laminectomy was performed at the thoracic 9 (T9) level to expose the spinal cord. A lateral hemisection was then performed using a micro knife (10315-12, Fine Science Tools, Heidelberg, Germany) under the microscope (Leica M80, Wetzlar, Germany), as described previously [32]. Immediately following injury, the viral vector was injected using a Hamilton microneedle (RN, gauge 29, 25 mm, 12°) with 10 µL Hamilton micro-syringe (both from Fisher Scientific, Illkirch, France) connected to UltraMicroPump III microinjector (World Precision Instrument, Sarasota, FL, USA). Injections were performed in four locations (Figure 1A), as previously described [33] (one injection point within the lesion site, one injection 1mm rostral to the lesion, one injection 1mm caudal to the lesion and one injection contralateral to the injury, respectively). For each injection, 0.5 µL of the vector was infused at 0.5 and 1 mm depth at a rate of 5 nL/s. Thus, the total volume of injected vector in the spinal cord was 4 µL. Following injections, muscles and skin were closed (vicryl, 3/0 and Filapeau 4/0, Braun, Germany, respectively). Animals were left to recover on a warming pad and monitored over 1-h after anesthesia before returning to their home cage. Manual bladder-voiding was performed twice a day until total recovery of sphincter control. Bodyweights were measured prior to and after surgery throughout the study. Animals were euthanized if signs of pain, infection, or weight loss higher than 20% were observed.

### 2.4. Behavioral Analysis

Habituation to the tests was carried out 2 weeks before surgery. Behavioral assessments were performed 1 week and 24 h prior to surgery to set pre-operative values. Then, behavioral analyses were carried out at 3 days, 5 days, 1 week, and once a week over 6 weeks after surgery.Open field: Animals were placed in a 50 × 50 cm square arena. Spontaneous motor activity was video-recorded over 10 min. The first two minutes were considered habituation time and were systematically excluded from analysis. Recordings were first performed on a smooth ground and then repeated on sandpaper surfaces with two types of granularities, i.e., 50 and 240 mean grain size/µm (adapted from [34,35]). The following parameters were analyzed: time spent by zone (s), speed by zone (cm/s), and time of immobility (s) (defined as no movement for more than 2 s). Zones were set as follow: total arena, arena center, arena periphery for smooth surface and zone of granularity 50, zone of granularity 240 for sandpaper surface. Center region size was set as 20 × 20 cm, and the periphery corresponds to the remaining area. Zones of granularities 50 and 240 were designed as 25 × 25 cm square and placed in alternance. Ethotrack software (Innovation Net, Tiranges, France) was used for the automatized video tracking and analysis.CatWalk^TM^: A dynamic walking pattern was analyzed using the CatWalk™ test (CatWalk XT™, Noldus, Wageningen, The Netherlands). Animals walked through a corridor on a backlighted glass plate. Paw placements were recorded by a camera placed under the glass plate. Six runs per sessions were recorded. Runs were analyzed only if the following criteria were met: average speed comprised between 5 and 30 cm/s, and combined to a maximum speed variation of 70% (adapted from [36]). Several parameters were analyzed, including base of support, print position, and max contact. Animals included in the experimental and control groups had similar weights and average motion speed, allowing for equivalent detection and comparison between groups, as previously described [36]. Additionally, we have quantified the percentage of detected ipsilateral hind paws prior traumatism and over the first week after SCI.

Number of mice included in the behavioral study: 18 C57BL6/6J underwent SCI and pHIV-Fgfr4 vector injections (12 females and 6 males, referred to as experimental mice); 19 mice underwent SCI and pHIV-mCherry vector injections (12 females and 7 males, referred to as control mice).

### 2.5. Tissue Processing

Animals were sacrificed at specific time points depending on the experiment (MRI and histology). Following peritoneal injection of tribromoethanol (500 mg/kg, Sigma-Aldrich, Darmstadt, Germany), blood was chased with an intracardiac perfusion of 4 °C 0.1 M phosphate buffer saline (PBS), followed by cold 4% paraformaldehyde (PFA) in 0.1 M PBS, pH 7.2. Spinal cord and gastrocnemius-soleus-plantaris muscular complexes were dissected and post-fixed in 4% PFA for 2 h. For ex vivo DW-MRI analysis, samples were stored in 1% PFA until acquisitions. For histology, samples were incubated in 30% sucrose in 0.1 M PBS, frozen in OCT (Sakura, Alphen aan den Rijn, The Netherlands) and stored at −20 °C.

### 2.6. Ex Vivo Diffusion Weighted Magnetic Resonance Imaging (DW-MRI)

Dissected spinal cords were placed in a glass tube filled with fluorinert FC-770A (3M™ Electronic Liquids, Saint Paul, MI, USA) within a solenoid antenna. The coil was specifically designed for spinal cord studies [37]. Acquisitions were performed with a 9.4 Tesla MRI (Agilent Varian 9.4/160/ASR, Santa Clara, CA, USA) associated with a VnmrJ acquisition system (Agilent, Palo Alto, CA, USA). Acquisition parameters were the following: spin-echo multi slice (SEMS, TR = 1580 ms TE = 30.55 ms, NE = 1, AVG = 30, FOV = 10 × 10, 36 slides, thickness = 1 mm, gap = 0 mm, acquisition matrix (NREAD*NPHASE = 128 × 128). Diffusion gradients were applied in three directions (Gs = 10 G/cm; delta = 6.844 ms; separation = 15.05 ms; b-value = 499.21 s/mm^2^). Regions of interest were manually segmented using Myrian software (Intrasens, Montpellier, France) and data were processed using a MATLAB-based in-house toolbox, as we previously described [34]. The following parameters were analyzed: lesion area at the epicenter, lesion extension, and lesion volume. Number of C57BL6/6J mice included in the MRI study: 18 C57BL6/6J experimental mice (12 females and 6 males); 19 control mice (12 females and 7 males). Animals were sacrificed at 6 weeks after SCI.

### 2.7. Immunohistochemistry

Frozen spinal cords were cut either transversally (14 µm thick) or longitudinally (22 µm thick) using a Cryostat (Microm HM550, Thermofisher Scientific, Waltham, MA, USA).Immunoperoxidase: Sections were washed in 0.1 M PBS and incubated for 15 min in hydrogen peroxide solution (H_2_O_2_, 1% in 0.1 M PBS, Sigma Aldrich, Gilligham, UK), washed in 0.1 M PBS for 10 min and incubated in lysine 20 mM, pH 7.4 for 20 min. After two washes in 0.1 M PBS, sections were incubated in blocking buffer (BSA 1%, triton ×100 0.1%, PBS 0.1 M) for 2 h, at room temperature (RT). Sections were then incubated for 48 h at 4 °C with the primary antibody, washed three times in 0.1 M PBS and incubated in the corresponding secondary antibody for 2 h at RT. Then, sections were washed with 0.1 M Trizma Base Saline (TRIS) for 3 × 10 min and the peroxidase reaction product was revealed using DAB (3,3′-Diaminobenzidine) substrate kit (Vector Labs, Burlingame, CA, USA). The reaction was stopped by washing sections in 0.1 M TRIS for 3 × 10 min. Sections were then dehydrated with increasing concentrations of ethanol (70, 80, 90, and 100%), cleared using xylene and coversliped (Eukitt**^®^** Mounting Medium, Sigma Aldrich, Gilligham, UK). To limit bias due to experimentation, staining of all sections was conducted simultaneously for a given protein and time point.Immunofluorescence: Sections were washed in 0.1 M PBS, incubated in lysine 20 mM, pH 7.4 for 20 min. After two washes in 0.1 M PBS, sections were incubated in blocking buffer (BSA 1%, triton ×100 0.1%, PBS 0.1 M) for 2 h at RT. Sections were then incubated for 24 or 48 h at 4 °C with the primary antibody, washed three times in 0.1 M PBS and incubated in the corresponding secondary antibody (2 h, RT). Sections were washed 3 × 10 mn in 0.1 M PBS and coversliped using fluorescent mounting medium (Dako, Glostrup, Denmark). Fluorescent sections were kept away from light and stored at 4 °C.Myelin staining: we used fluoromyelin as previously described in [38,39]; in short, sections were rinsed in 0.1 M PBS for 1 min and then incubated in FluoroMyelin™ (1:200, Thermofisher Scientific, Waltham, MA, USA) for 20 min at RT. Sections were washed 3 × 10 mn in 0.1 M PBS, coversliped using fluorescent mounting medium (Dako, Glostrup, Denmark), were kept away from light, and stored at 4 °C.Neuromuscular junctions: staining on transverse cryosection (16 µm) of the gastrocnemius–soleus–plantaris muscular complex was carried out (Microm HM550, Thermofisher Scientific, Waltham, MA, USA) using enzymatic method [40]. Sections were washed in 0.1 M PBS and incubated for 30 min in a solution containing: 0.5% acetylthiocholine (Sigma Aldrich, Saint Louis, MO, USA), 0.1 M sodium acetate pH 6, 0.1 M sodium citrate pH 6, 30 mM CuSO_4_, and 5 mM potassium ferricyanide. Sections were then washed in 0.1 M PBS and dehydrated with increasing concentrations of ethanol (70, 80, 90, and 100%) followed by xylene before applying coverslips (Eukitt**^®^** Mounting Medium, Sigma Aldrich, Gilligham, UK).Antibodies: Primary antibodies: rat anti mCherry monoclonal antibody 16D7 (1:500, Invitrogen, Carlsbad, CA, USA), rabbit anti GFAP (1:1000, Dako, Glostrup, Denmark), rabbit anti Iba1 (1:1000, Wako Pure Chemical Industries, Osaka, Japan), rabbit anti GAD65/67 (1:500, Abcam, Cambridge, UK), rabbit anti GAP 43 (1:1000, Milliport, Dramstadt, Germany), and mouse anti βIII-tubulin (1:100; MAB1195; R&D Systems, Minneapolis, MS, USA).Secondary antibodies: donkey anti rat alexa 594 (1:1000, Invitrogen, Carlsbad, CA, USA), APC-conjugated donkey anti mouse (1:100; Invitrogen, Carlsbad, CA, USA), donkey anti rabbit peroxidase (1:500, Jackson Immunoresearch, Stratech Scientific Ltd**.**, Soham, UK), and goat anti-mouse alexa 350 (1:1000, Invitrogen, Carlsbad, CA, USA).

### 2.8. Microscopy and Quantifications

All quantifications were carried out blindly.Immunohistochemistry: Transverse sections were scanned using NanoZoomer RS slide scanner (NanoZoomer Digital Pathology System and NDP view software, Hamamatsu City, Japan). All images were acquired at the same light exposure and exported with identical parameters. Optical density (OD) was measured using Image J software (National Institutes of Health, Bethesda, MD, USA). Background (OD without tissue) was subtracted for each section. Number of mice included in the study: 5 C57BL6/6J female experimental mice; 6 C57BL6/6J female control mice. Animals were sacrificed at 6 weeks after SCI.Quantifications of vector transduction in GFAP positive astrocytes: longitudinal sections of spinal cords were imaged with upright fluorescence microscope Axio Imager M1 (Zeiss, Oberkochen, Germany) at 1020 µm, 2040 µm, and 3060 µm, rostral and caudal to the lesion site. Settings were kept constant for all acquisitions. GFAP^+^ cells, mCherry^+^ cells, and GFAP^+^/mCherry^+^ cells were manually quantified using Multi-Point tool in Image J software (). Number of mice included in the quantifications: three injured male mice injected with pHIV-mCherry. Animals were sacrificed at 2 weeks after SCI.Quantifications of spared myelin: images of 14 µm axial spinal cords cryosections were acquired with THUNDER Imager 3D (Leica, Wetzlar, Germany; lens × 63) 3150 µm, rostral and caudal to the lesion site. One field of 600 µm × 400 µm was acquired in both lateral funiculi and 3 images of 40 µm × 40 µm located dorsal, central, and ventral to the spinal cord were taken for quantification. Spared myelinated fibers were quantified using multi-point tool of ImageJ software (Rasband, W.S., ImageJ, U. S. National Institutes of Health, Bethesda, MD, USA, https://imagej.nih.gov/ij/, 1997–2018) with numeric zoom to reach 200% of the original image. Number of mice included in the study: 6 C57BL6/6J female experimental mice; 6 C57BL6/6J female control mice. Animals were sacrificed at 6 weeks after SCI.In Figure 2, fluorescent THUNDER Imager 3D (Leica, Wetzlar, Germany; lens × 40) images were taken rostral and caudal to the lesion. Confocal images: sections were imaged with a laser scanning confocal microscope (Leica SPE, Mannheim, Germany) associated with a Leica LAS AF interface. Settings were kept constant for all acquisitions.

### 2.9. Fluorescence-Activated Cell Sorting (FACS)

Mice were sacrificed at 2 weeks after surgery with a lethal dose of tribromoethanol (i.p., 500 mg/kg, Sigma-Aldrich, Darmstadt, Germany). The blood was washed out through intracardiac perfusion with cold 0.1 M PBS. Spinal cords were dissected, washed in 0.1 M PBS and transferred in 750 uL of PBS. Mechanical trituration was followed by enzymatic tissue digestion through 30 min incubation at 37 °C in the following solution: hyaluronidase 7 mg/mL, trypsin 13 mg/L, kynurenic acid 4 mg/mL (all from Sigma Aldrich, Saint Louis, MS, USA), DNAse I 10 mg/mL (Roche, Rotkreuz, Switzerland), and Mg2^+^ 25 mM (Sigma Aldrich, Saint Louis, MO, USA). Single-cell suspension was filtered on a 40 µm sieve (BD Biosciences, Franklin Lakes, NJ, USA), the sieve was washed with 0.1 M PBS and the solution was centrifuged at 700× *g* for 5 min. Supernatant was resuspended using 0.9 M sucrose in PBS and centrifuged at 700× *g* for 20 min to discard myelin fragments. Cells were incubated in mouse anti βIII-tubulin in PBS (1:100; R&D Systems, Minneapolis, MS, USA) for 20 min on ice. Cells were centrifuged for 5 min at 400× *g*, washed with cold 0.1 M PBS and incubated for 15 min on ice in Allophycocyanin (APC)-conjugated donkey anti mouse in PBS (1:100; Jackson Immunoresearch, Carlsbad, CA, USA). As controls, we used wild type mice (to show cell distribution without any staining), uninjured mice (to show eGFP expression alone), wild-type mice stained with βIII-tubulin (to show APC staining alone), and without primary antibody (Appendix A). Cells were centrifuged (5 min at 400× *g*), washed with cold 0.1 M PBS and re-suspended in 7-AAD before sorting. We used BD FACSAria™III. (Becton Dickinson cell sorter BD Biosciences, Franklin Lakes, NJ, USA) equipped with 638 nm laser (670-14) and 561 nm laser (610-20) and controlled using FACSDiva software (Becton Dickinson cell sorter BD Biosciences, Franklin Lakes, NJ, USA). Size threshold was used to eliminate debris. Two cell populations were sorted simultaneously: non-converting (Aldh1l1eGFP^+^/βIII-tubulin^−^) and converting astrocytes (Aldh1l1eGFP^+^/βIII-tubulin^+^). Number of mice included in the FACS experiment: 30 Aldh1l1-EGFP hemisected male mice.

### 2.10. RNA-Seq ANALYSIS

A total of 1400 converting (Aldh1l1eGFP^+^/βIII-tubulin^+^) and 4000 non-converting (Aldh1l1eGFP^+^/βIII-tubulin^−^) astrocytes were used to directly generate full-length cDNA (SMART-Seq kit v4 Ultra Low Input RNA Kit for Sequencing, Clontech Laboratories, Inc., Mountain View, CA, USA). Cells were obtained from three independent experiments each having at least five injured mice. Cells were collected in 9 µL Rnase-free 0.1 M PBS. Then, 1 μL of 10× reaction buffer was added into each sample and cells were transferred to a 0.2-mL Rnase-free PCR tube. Cells were then incubated for 5 min at RT and placed on ice. Following, 2 μL of 3′ SMART-Seq CDS Primer II A was added and mixed well by gently vortexing. Samples were incubated at 72 °C in a pre-heated, hot-lid thermal cycler for 3 min and placed on ice. Then, 4 µL of 5× Ultra Low First-Strand Buffer, 1 µL of SMART-Seq v4 Oligonucleotide (48 μM), 0.5 µL of Rnase Inhibitor (40 U/μL), and 2 μL of the SMARTScribe Reverse Transcriptase were then added to each sample before placing them in a thermal cycler at 42 °C for 90 min followed by 70 °C for 10 min. Sequencing quality control was performed using FastQC v.0.11.2. The reads containing adapter sequences were removed with FASTX-Toolkit. The reads were mapped with the TopHat v2.0.13 software to the reference genome and on new junctions with known annotations. Biological quality control and summarization were carried out with RseQC v2.6.1 and the PicardTools v1.80. The percentage of mRNA base enrichment ranged between 60 and 89%. Data normalization and differential expression analysis was performed using R/Bioconductor edgeR software v.3.10.5 for the genes annotated in the reference genome. Data were normalized according to library size. To identify transcripts that were differentially expressed, we defined a criterion of a two-fold and greater difference plus significant (*p* < 0.05) *p* value. Statistics: *t*-test with un-equal variance. Pathway analysis was carried out using MetaCore, as we previously described [41].

### 2.11. Statistics

Behavioral experiments: recovery was assessed using two-way repeated measure ANOVA followed by Bonferroni post hoc tests. Paired *t*-test was used for comparison between ipsilateral and contralateral sides. Unpaired *t*-test with Welch’s correction was used for all other comparisons. Significance was accepted at *p* ≤ 0.05. Results are expressed as mean ± standard error of the mean (SEM). All statistical analyses were performed using GraphPad Prism software.

## 3. Results

### 3.1. Lentiviral Mediated Over-Expression of Fgfr4 in Astrocytes Persists after SCI and Is Associated with βIII-Tubulin Expression

With the aim of increasing spontaneous astrocyte-to-neuron conversion after SCI, we over-expressed Fgfr4 specifically in astrocytes post-injury using a lentiviral vector-mediated approach. We selected Fgfr4 for its potential as an endogenous modulator/inductor of astrocyte-to-neuron conversion. Indeed, we have previously identified that SCI induced an up-regulation of Fgfr4 mRNA in reactive astrocytes and that the peak of FGFR4 protein expression preceded the peak of βIII-tubulin expression [13]. In addition, FGFR4 promotes differentiation of embryonic stem cells towards neuronal lineage [15]. As a pre-requisite, we first verified the specific targeting and transduction efficiency of astrocyte within the injured spinal cord using a control lentiviral vector that encodes mCherry under the GFAP promotor (pHIV pGFAP mCherry; hereafter referred ti as pHIV-mCherry) (Figure 1A). We assessed astrocytic targeting using immunohistochemistry in C57BL6/6J adult mice. Three mice underwent T9 lateral hemisection of the spinal cord and were injected with pHIV-mCherry (Figure 1B) immediately post lesion, as previously described [33].

Two weeks after the injury, immunohistochemistry on longitudinal spinal cord sections allowed the identification and quantification of astrocytes (GFAP^+^ cells) as well as transduced astrocytes (mCherry^+^/GFAP^+^). Quantifications were performed at 1020 µm, 2040 µm, and 3060 µm distances rostro-caudal to the lesion epicenter (Figure 1C). mCherry^+^/GFAP^+^ cells were detected over 3060 µm rostral and caudal to the lesion site (Figure 1D–E), suggesting that astrocytic transduction extends over a large distance across the lesion site. Quantification of transduced astrocytes (GFAP^+^/mCherry^+^) highlighted an overall mean of 12.82% ± 4.88 of astrocytic transduction within the 6 mm segment. However, in close vicinity to the lesion, 13.17% ± 5.5 and 21.69% ± 10.94 (1020 µm rostral and caudal to the lesion, respectively) of the astrocyte were transduced (Figure 1E). We then assessed GFAP/mCherry co-expression 6 weeks after SCI using another group of mice and demonstrated that astrocyte transduction persists at chronic stages post-SCI (Figure 1F–H, arrowheads in H).

We have previously identified that SCI induces the expression of βIII-tubulin in 10% of resident reactive astrocytes surrounding the lesion site. Thus, in another group of mice, we injected a lentiviral vector expressing Fgfr4 under the GFAP promotor that also encodes mCherry as a reporter protein (pHIV pGFAP Fgfr4 mCherry, hereafter referred to as pHIV-Fgfr4) and examined whether astrocytes expressing lentiviral mediated Fgfr4 also expressed βIII-tubulin at 6 weeks post-SCI. Fluorescent (THUNDER Imager, Leica, Figure 2A–H) and confocal (Figure 2I–L) acquisition were carried out 2mm rostral to the lesion (Figure 2A–H) and in the vicinity of the lesion (Figure 2I–L). Some astrocytes (Figure 2E,I) transduced by pHIV-Fgfr4 (Figure 2F,J) that indeed expressed βIII-tubulin (Figure 2G,K,H,L, arrowheads in H,L) were observed.

Taken together, our results demonstrate that lentiviral-mediated mCherry expression in astrocytes persists 6 weeks after SCI. Moreover, lentiviral-mediated astrocytic Fgfr4 over-expression is associated with βIII-tubulin expression.

### 3.2. Lentiviral-Mediated Over-Expression of Fgfr4 in Astrocytes Improves Functional Recovery after SCI in a Gender-Dependent Manner

We then investigated the functional consequences of astrocytic Fgfr4 over-expression immediately after SCI. Male and female C57BL6/6J mice underwent the same protocol as described above (Figure 1B). The experimental group received pHIV-Fgfr4 whilst the control group was injected with pHIV-mCherry. For all tests and for each animal, behavior data were normalized to values obtained prior to SCI (Figure 3, dashed lines, 100%). We used two behavioral tests over a 6-week period following the lesion. First, we examined gross motricity using open field. Female mice in the experimental group showed increased gross motor function as evidenced by greater distance walked as compared to the control group throughout the 6 weeks post-SCI (Figure 3A). Specifically, pHIV-Fgfr4 female mice covered a distance closer to the value obtained prior to injury than pHIV-mCherry-injected animals. Conversely, no difference between groups was observed in male mice (Figure 3B). Other parameters such as the time spent in the arena center square (Appendix A) and the time of immobility, defined as no movement for more than 2 s (Appendix A), did not differ between groups in both sexes. We also examined gross tactile sensitivity through the quantification of the time spent on two granularities of sandpaper [35], which showed no difference between groups in both sexes (Appendix A). Next, we assessed fine motricity and motor coordination using CatWalk™ analysis. Due to SCI, the ipsilateral hind paw could not be detected before the second week following injury in both groups and sexes (Figure 3E,F, black insets). However, as the lack of detected print reflects the absence of ipsilateral hindpaw placement, we have analyzed over the first week after SCI the detection percentage of the ipsilateral hindpaw (Figure 3C–D). No significant difference in both groups and sexes was observed over the first week after injury.

The “print position” ipsilateral to the lesion (representing the distance between the placement of the hind paw and the placement of the front paw over a step cycle) highlighted a better recovery for pHIV-Fgfr4-injected females as compared to the control group (Figure 3C). Indeed, pHIV-Fgfr4-injected females displayed print position values closer to those obtained prior to SCI ipsilateral (Figure 3E). In contrast, no difference between the groups was observed in males (Figure 3F). The “print position” contralateral to the lesion (Figure 3E) showed no difference between groups in both sexes (Figure 3G–H). The “max contact time” (reflecting the time of maximum contact with the plate during the transition from braking to propulsion phase) of the contralateral hind paw were similarly affected between control and experimental groups in female mice (Figure 3I). Conversely, we quantified a better recovery in pHIV-Fgfr4-injected males as compared to the control group (Figure 3J), as reflected by values closer to those obtained prior to SCI. The base of support that corresponds to the distance between the placement of either the front (Appendix A) or the hind paws (Appendix A) did not differ between groups for both sexes.

Taken together, our results demonstrate that astrocytic Fgfr4 over-expression immediately after SCI improves functional motor recovery. Moreover, enhanced recovery may reflect a sexual dimorphism. Indeed, pHIV-Fgfr4 injection improves gross motricity only in females and fine motor functions in a sex-dependent manner.

### 3.3. pHIV-Fgfr4 Condenses Lesion Volume Specifically in Female Mice after SCI

Six weeks after injury, to investigate whether astrocytic Fgfr4 over-expression affects spinal cord lesion at tissue level, we analyzed the lesion volume using high resolution ex vivo diffusion magnetic resonance imaging (DW-MRI). We manually segmented the lesion area on 1 mm-thick slices throughout 2 cm-spinal cord segment centered on the lesion site (Figure 4A–F). We then examined lesion area at the epicenter (Figure 4G,H), lesion extension (Figure 4I,J), and lesion volume (Figure 4K,L). Lesion extension, and lesion volume were reduced in the pHIV-Fgfr4-injected females (Figure 4I,K) as compared to the control group. In contrast, no difference in any of the parameters was observed between male groups (Figure 4H,J,L).

To further analyze the effect of pHIV-Fgfr4 injections after SCI on tissue sparing in female mice, we investigated myelin fibers integrity using fluoromyelin staining 6 weeks after SCI (Figure 5). Fibers with complete myelin sheets (Figure 5B–F, arrow) were quantified using axial sections located at 3 mm distance rostral and caudal to the lesion epicenter. Fibers with incomplete myelin sheets were excluded (Figure 5C–E, arrowhead). In each axial section, six high magnification (×63) images were taken within the lateral columns (Figure 5A) in pHIV-mCherry (Figure 5C,E), and pHIV-Fgfr4 (Figure 5D,F) animals. Quantification of myelin fibers revealed significant reduction in spared myelin fiber density in pHIV-mCherry mice as compared to uninjured animals; conversely, no significant difference was observed between uninjured and pHIV-Fgfr4 groups (Figure 5G). Comparison of spared myelin fibers density between ipsilateral and contralateral sides highlighted no difference in pHIV-mCherry mice in contrast with the pHIV-Fgfr4 group, which presented a higher density on the contralateral side of the lesion (Figure 5H).

Taken together, these data suggest that a post-SCI enforced Fgfr4 over-expression in astrocytes leads to a condensed lesion size and an overall decrease in secondary damage on myelinated fibers.

### 3.4. pHIV-Fgfr4 Reduces Glial Reactivity in Female Mice after SCI

To better examine the histological outcome upon astrocytic Fgfr4 over-expression in female mice after SCI, we used quantitative immunohistochemistry to analyze astrocyte reactivity in the control (Figure 6A–C), and experimental (Figure 6D–F) groups.

Only females were analyzed since treated mice displayed an overall better functional recovery and condensed lesion size than males. GFAP expression in the grey and the white matters was quantified on 5 mm segments rostral (Figure 6G,H), and caudal (Figure 6I,J) to the lesion. In the rostral segment, no difference was observed between groups in the grey (Figure 6G), and the white (Figure 6H) matters. Conversely, caudal to the lesion, astrocytic reactivity was reduced in the experimental group as compared to the control in both grey and white matters (Figure 6I,J). Next, we analyzed microglia reactivity in the control (Figure 6K–M) and experimental (Figure 6N–P) groups. Rostral to the lesion, IBA1 expression was reduced only in the grey matter (Figure 6Q). No difference was observed in the white and the grey matters between groups in the caudal segments (Figure 6S,T). Taken together, our results demonstrate that post-injury Fgfr4 over-expression in astrocytes limits astrocytic reactivity caudal to the lesion and microglial response rostral to the lesion in female mice.

### 3.5. Positive Effect of pHIV-Fgfr4 on Motor Recovery after SCI Is Not Reflected by Muscles’ Surface and Neuromuscular Junction Density

Six weeks after injury, to investigate whether an increased astrocytic expression of Fgfr4 could preserve the muscular component, we analyzed the gastrocnemius-soleus-plantaris complexes of both hind limbs (Figure 7A). No difference either in the surface area nor neuromuscular junction density was observed between groups (Figure 7B–D).

Taken together, our results demonstrate that Fgfr4 over-expression in astrocytes immediately after SCI improves functional motor recovery without modifying neuromuscular junction density.

### 3.6. Molecular Signature of SCI-Induced βIII-Tubulin-Expressing Astrocytes Is Consistent with Their Transdifferentiation State into Neuron-like Cells

Finally, to initiate investigations into molecular mechanisms underlying SCI-induced astrocytic transdifferentiation, we took advantage of the Aldh1l1-EGFP transgenic mice that express enhanced green fluorescent protein (eGFP) in astrocytes. Aldehyde dehydrogenase 1 family member L1 is a specific pan-astrocytic marker [7]. We combined fluorescence-activated cell sorting (FACS) and RNA-sequencing (RNA-Seq) analysis. We chose βIII-tubulin as a marker of astrocyte-to-neuron conversion since following SCI 10% of reactive astrocytes co-express βIII-tubulin [13]. Two weeks after SCI, using FACS, we simultaneously isolated non-converting (Aldh1l1eGFP^+^/βIII-tubulin^−^) and converting astrocytes (Aldh1l1eGFP^+^/βIII-tubulin^+^) (Appendix A). To overcome limited RNA input, we used the SMART-SeqTM technology, which generates full-length cDNAs and good read coverage across the whole transcript [for review see [42]]. We then carried out cell-specific RNA-Seq analysis to compare molecular signatures of converting and non-converting astrocytes originating from the same injured spinal cords. We selected 2 weeks after spinal cord hemisection since 14.96% of astrocytes expressed βIII-tubulin protein whilst uninjured spinal cord does not show any Aldh1l1eGFP^+^/βIII-tubulin^+^ co-expressing cells [13]. Using 2-fold difference (logFC = 1) and *p* value ≤ 0.05, we identified over 284 differentially expressed genes between converting and non-converting astrocytes; 212 (75%) were down-regulated and 72 (25%) were up-regulated (Appendix A). Gene ontology (GO) analysis of down-regulated genes identified a significant enrichment of genes involved in “signal transduction and Notch signaling” (process network ranked 7th, P = 1.28 × 10^−2^), in particular through the down-regulation of Egfr (epidermal growth factor receptor, logFC = −9.69), Dlk (delta like non-canonical Notch ligand 1, log FC = −8.14), Fzd8 (frizzled class receptor 8, logFC = −6.89), and Pdgf-R (receptor platelet derived growth factor receptor, logFC = −8.53). Moreover, we identified a significant enrichment in the process network “development and neurogenesis and synaptogenesis” (ranked 11th, P = 2.345 × 10^−2^), in particular through the up-regulation of Wnt5b (wingless-type MMTV integration site family, member 5B, logFC = 10.640), and Kallikrein 7 (also known as neurosin, and Klk7, logFC = 9.251) (Appendix A).

Taken together, these results highlight an overall down-regulation of Notch signaling associated with an up-regulation of neurogenic process and therefore reinforce the hypothesis that βIII-tubulin-expressing astrocytes are indeed engaged in neuronal conversion.

## 4. Discussion

Previously, we had shown that SCI induces an increase in astrocytic Fgfr4 expression that precedes their phenotypic conversion toward neuronal lineage [13]. To examine its therapeutic potential, in this study we first show that viral vector-based Fgfr4 over-expression in astrocytes immediately after SCI improves functional motor recovery in a sex-dependent manner in adult mice. Better functional recovery in female mice is associated with condensed lesion size and reduced glial reactivity. Finally, gene pathways analysis indicates that SCI-induced astrocyte-to-neuron conversion involves a concomitant down-regulation of Notch signaling and an up-regulation of the neurogenic process.

Spontaneous and enforced glial conversion into neuron is a promising therapeutic approach to limit neuronal death and protect tissue structure to eventually restore altered functions in diseased and/or traumatized CNS (reviewed in [43]). In particular, achievements of direct in vivo reprograming of glial cells after traumatic CNS injuries had been shown (extensively reviewed in [44]). The first study converting reactive glial cells into immature neurons was performed after cortical stab wound injury and demonstrated the key role of oligodendrocyte transcription factor (Olig2) in repressing neurogenesis in reactive cells after lesion [45]. Reprograming of reactive astrocytes into functional neurons after stab wound brain injury had been achieved using retroviral delivery of NeuroD1. Indeed, three weeks after viral delivery, transduced astrocytes expressed neuronal nuclei (NeuN) and displayed extensive neurites [46]. Retroviral-mediated expression of the transcription factor (sex-determining region Y)-box2 (Sox2) induced the conversion of neurogenin2 positive cells (NG2^+^) to doublecortin positive neurons (DCX^+^) following cortical stab wound injury in adult mice [47]. AAV-based therapy has been recently developed to express NeuroD1 in reactive astrocytes after local ischemic brain injury [48]. A large number of astrocyte-to-neuron conversion was induced that generated functional neurons displaying axonal projections to distant targets and was associated with partial rescue of motor and fear memory deficits [48]. Similarly, AAV-mediated NeuroD1-delivery post-SCI converted reactive astrocytes into functional neurons that could generate action potential and integrate in the local synaptic network [30]. Indeed, following T11–T12 stab injury of the spinal cord, an in-depth characterization of NeuroD1-converted neurons proved their glutamatergic identity [30]. Likewise, AAV-mediated NeuroD1-delivery 10 days and 16 weeks after T11–T12 moderate contusion injury of the spinal cord induced astrocytes-to-neuron conversion by 10 weeks post injection [30]. However, in both SCI models, the authors did not investigate the functional and histological impacts of NeuroD1-neurons on the injured spinal cord. Importantly, it had been recently demonstrated that AAV-mediated expression of NeuroD1 is not restricted to astrocyte [49]. Lentiviral-mediated Sox2-delivery reprogramed adult spinal cord astrocytes to neurons after T8 lateral hemisection. Indeed, 3–6% of resident astrocytes converted to DCX^+^/βIII-tubulin^+^ co-expressing neuroblasts that gives rise to mature neurons [28]. In a follow-up investigation using T8 moderate contusion, the p53-dependent pathway had been identified as a key regulator for Sox2-mediated reprograming of astrocytes to neurons [29]. Finally, lentiviral-mediated Zfp521-delivery 1 week after T10 spinal cord contusion reprogramed astrocytes into βIII-tubulin^+^ cells, improved functional recovery, and motor evoked potential in adult rats [31]. To the best of our knowledge, our study is the first to show that an improvement in motor function recovery can be induced through the reinforcement of a molecular mechanism that spontaneously occurs in reactive astrocytes after SCI. Six weeks after injury, pHIV-Fgfr4 gene therapy targeting astrocytes led to a decrease in lesion volume accompanied by a reduced astrocytic (and to a lesser extent microglial) reactivity. Reduced astrocytic reactivity was particularly evident caudal to the lesion where we observed greater number of fgfr4-transduced astrocytes. This is consistent with previous studies demonstrating that blocking FGF signaling triggers increased astrocytic reactivity as reflected by an up-regulation of GFAP expression and a hypertrophic morphology (reviewed in [50]). Astrocytes regulate synaptic formation, maturation, optimal function, and elimination (reviewed in [51]); however, our results indicate that pHIV-Fgfr4 gene therapy targeting astrocytes does not modify neither gastrocnemius–soleus–plantaris muscular complex muscles volume ipsilateral to the lesion nor neuromuscular junction density. However, it is possible that pHIV-Fgfr4 modifies synaptic function and/or repartition of muscle fiber types that may explain better motor function recovery following SCI.

We uncovered that pHIV-Fgfr4-induced improvement in motor function recovery may display a sexual dimorphism. Astrocytes from both females and males respond to gonadal hormones through their expression of estrogen, progesterone, and androgen receptors. Several studies have shown that estrogen and progesterone treatments reduce astrocytic reactivity in pathological conditions (for review see [52]). Sexual dimorphic differences in the inflammatory response induced by lipopolysaccharide have been reported, such as greater IL-1β [53,54], IL6, and TNFα [54] production in male than in female astrocytes. More recent data also suggest a sex-dependent difference in astrocytic phagocytosis and inflammation response upon insulin-like growth factor-I signaling [55].

C57BL/6J mice also display sexual dimorphic differences in astrocytic response following traumatic brain injury, with divergent findings [56,57]. Villapol and colleagues (2017) reported increased astrogliosis in males but not females at 1 day after moderate-to-severe controlled cortical impact injury [57]. In contrast, Jullienne and co-workers (2018) reported increased astrocytic reactivity and a lower astrocytic hypertrophy in females compared to males at 1 day following moderate controlled cortical impact injury [56]. Therefore, our results confirm a sexual dimorphism of astrocytic responses in C57BL/6J mice following SCI.

We highlighted that a lentiviral-mediated overexpression of astrocytic Fgfr4 is linked to a persistent expression of βIII-tubulin in astrocytes undergoing conversion towards neural lineage. We identified a concurrent transcriptomic down-regulation of Notch signaling and an up-regulation of neurogenic pathways in converting as compared to non-converting astrocytes. This is in agreement with several studies demonstrating that neuronal differentiation depends on the down-regulation of Notch signaling (reviewed by [50]). In particular, reducing Notch signaling is essential for the activation of a neurogenic program in astrocytes in a physiological context. Moreover, activation of Notch pathway in astrocytes reduces neurogenesis after stroke [58]. Deleting the Notch-mediating transcription factor Rbpj (recombination signal binding protein for immunoglobulin kappa j region) activates astrocyte neurogenesis in the striatum [59]. Strikingly, during sensory neurogenesis in the ophthalmic trigeminal placode, Notch down-regulation coincides with the peak of FGFR4 expression, and FGFR4-induced activation of FGF may act in parallel with Notch signaling to induce neurogenesis [60]. One working hypothesis is that pHIV-Fgfr4 gene therapy following SCI increases FGFR4-induced activation of FGF signaling via FGF2 since an up-regulation of FGF2 at the mRNA and protein level is observed after SCI (reviewed in [61]).

In vivo reprograming in SCI context is particularly challenging since different cell types are intertwined together and exposed to a highly dynamic damage environment. Here, we show that increasing an endogenous response of astrocyte following SCI promotes functional recovery and preserves spinal cord tissue in particular in adult female mice. These results further emphasize the importance of in vivo reprograming as a gene therapy strategy to promote recovery after SCI and stress that biological sex should be considered cautiously when designing new therapeutic approaches. In conclusion, our results suggest that gene therapy elevating an endogenously injury-induced response provides an interesting therapeutic approach after CNS traumatism.

## Figures and Tables

**Figure 1 cells-12-00528-f001:**
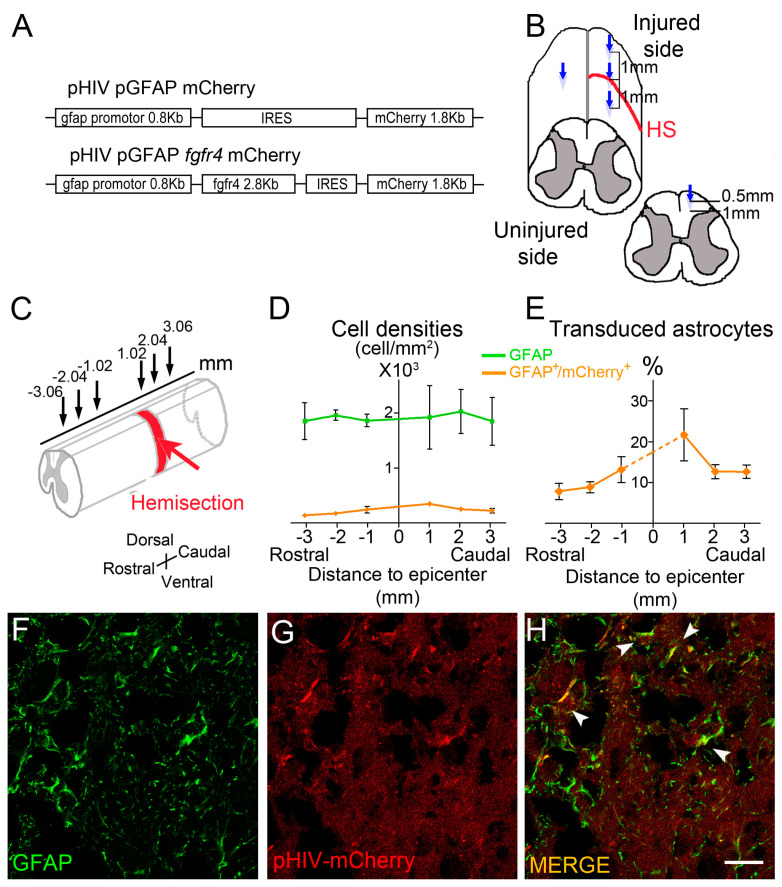
In vivo transduction of astrocytes after SCI: Schematic view of the vector constructs (**A**). Immediately following T9 lateral hemisection of the mouse spinal cord, 1 µL of vectors (pHIV-Fgfr4 or pHIV-mCherry) was injected into four distinct locations: within the lesion site, contralateral to the lesion site, 1 mm rostral, and 1 mm caudal to the lesion site. Each injection location consisted of two depths (0.5 and 1 mm depth in the dorsoventral axis) (**B**). To assess lentiviral vector diffusion and astrocytic transduction efficiency, immunohistochemistry analyses were performed 2 weeks after SCI (pHIV-mCherry-injected animals) in three distinct locations both rostral and caudal to the lesion (±1020 µm, ±2040 µm, and ±3060 µm, respectively) (**C**). Quantification of the density of GFAP^+^ cells (astrocytes) and mCherry^+^/GFAP^+^ cells (transduced astrocytes) (**D**). Percentage of transduced astrocytes (mCherry^+^/GFAP^+^ cells) relative to total GFAP^+^ population (**E**). Confocal images taken in close vicinity of the lesion 6 weeks after SCI (**F**–**H**). Transduced astrocytes persist at chronic stages (arrowheads, (**H**)). Three mouse spinal cord were quantified. Data are expressed as mean ± SEM. Scale bar: 20 µm.

**Figure 2 cells-12-00528-f002:**
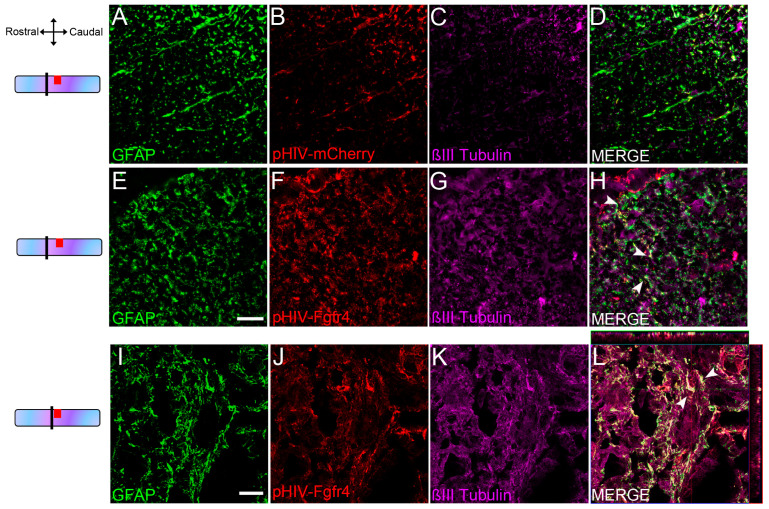
Transduced astrocytes express βIII-tubulin 6 weeks after SCI. Schemes on the left indicate the location of images. Representative THUNDER images of pHIV-mCherry-injected female mice (dorsal white matter) 2 mm rostral to the lesion at 6 weeks after SCI. Displayed images represent maximum intensity projection of a z stack of 10 planes with a 3 μm thickness (**A**–**D**). Representative THUNDER images of pHIV-Fgfr4-injected female mice (dorsal white matter) 2 mm rostral to the lesion at 6 weeks after SCI (**E**–**H**). Note the increase in βIII-tubulin expression in pHIV-Fgfr4-injected female mice compared to pHIV-mCherry-injected group (**A**–**D**). Representative confocal images of pHIV-Fgfr4-injected female mice in close vicinity of the lesion site, 6 weeks after SCI (**I**–**L**). Micrographs confirm βIII-tubulin protein expression in a sub-population of transduced astrocytes (arrowheads, (**H**)). Scale bars: 20 µm.

**Figure 3 cells-12-00528-f003:**
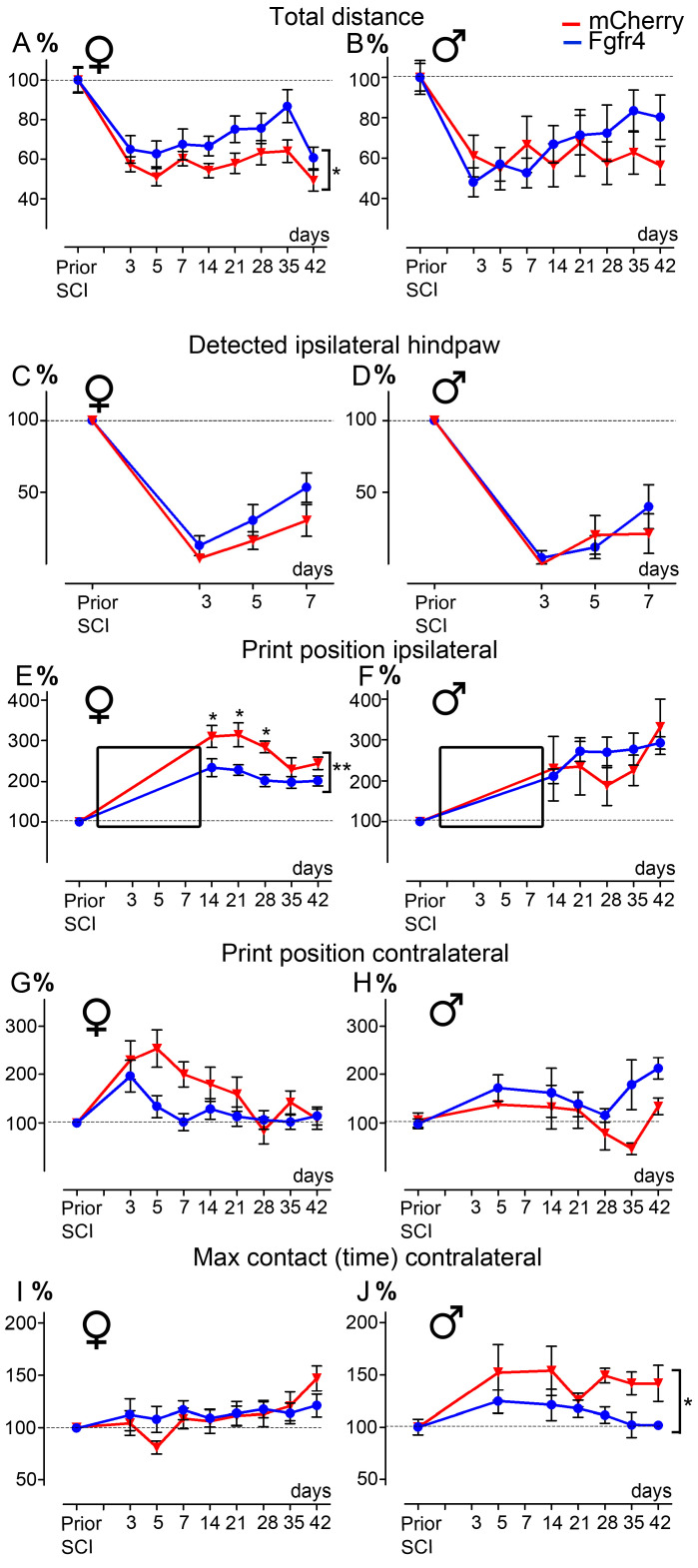
pHIV-Fgfr4 vector injection immediately after SCI improves functional recovery differentially in female and male mice. Open field graphs displaying the total distance covered by females (**A**), and males (**B**). Graph showing detected ipsilateral hindpaw using Catwalk^TM^ in females (**C**), and males (**D**) during the first week after SCI (correspond to black insets in **E** and **F** respectively). CatWalk^TM^ graphs showing the print position ipsilateral (**E**,**F**) and contralateral to the lesion (**G**,**H**) and the time of max contact of the hind paw contralateral to the lesion (**I**,**J**). Female (**A**,**C**,**E**,**G**,**I**) and male mice (**B**,**D**,**F**,**H**,**J**). All values were normalized to data obtained before SCI using the same animal (dashed line, 100%). In all graphs, animals that received injections of the experimental vector (pHIV-Fgfr4, blue) and the control vector (pHIV-mCherry, red) are represented. Data are expressed as mean ± SEM. Two-way ANOVA * *p* < 0.05, ** *p* < 0.01 (**A**,**C**,**H**) followed by Bonferroni post hoc test. Number of C57BL6/6J mice: 12 females and 6 males (pHIV-Fgfr4 vector) and 12 females and 7 males (pHIV-mCherry).

**Figure 4 cells-12-00528-f004:**
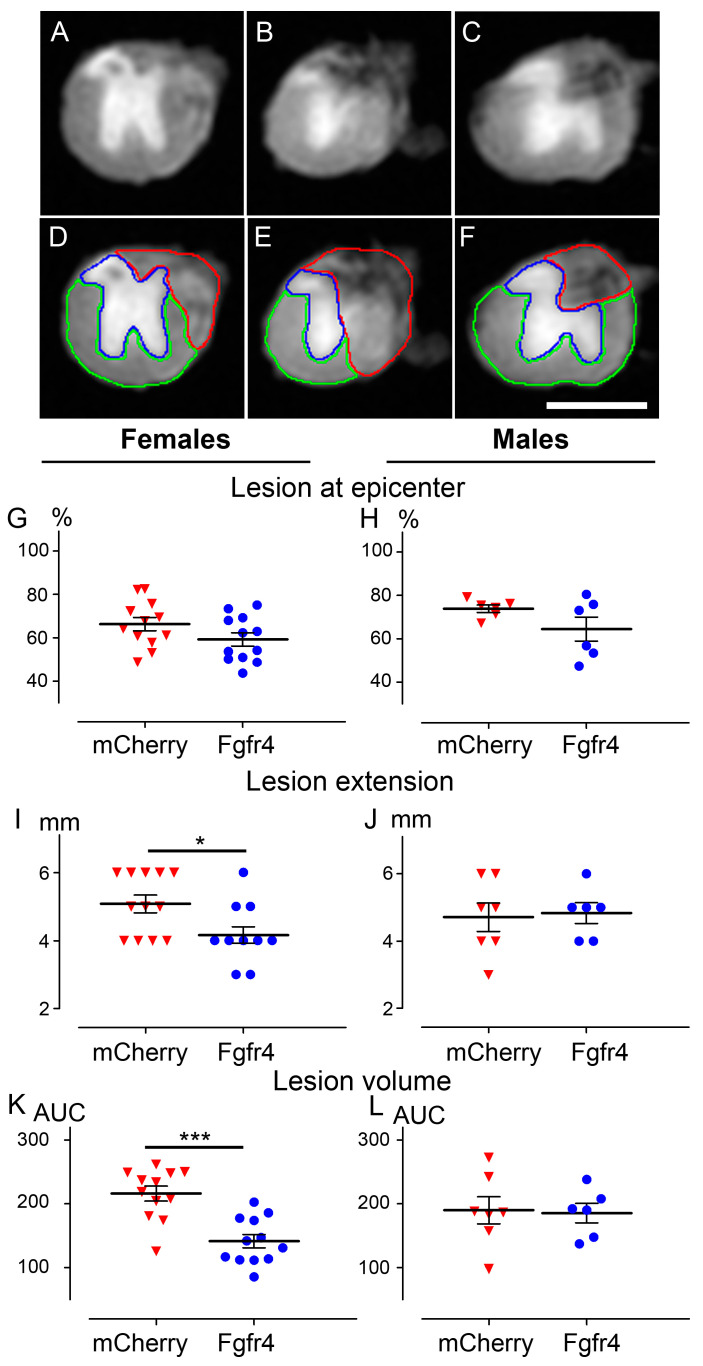
pHIV-Fgfr4 vector injection immediately after SCI preserves spinal cord tissues. Representative ex vivo axial DW-MRI (**A**–**F**) of the spinal cord from a C57BL6/6J female mouse that underwent injury and pHIV-Fgfr4 vector injections. Photograph within the lesion epicenter (**B**,**E**), 1 mm rostral (**A**,**D**), and 1 mm caudal (**C**,**F**) to the lesion site. Manual segmentations (**D**–**F**) of the spared grey matter (blue), the spared white matter (green), and the injured tissue (red). In females (**G**,**I**,**K**) and males (**H**,**J**,**L**), lesion area at the epicenter (**G**,**H**), lesion extension (**I**,**J**), and lesion volume (**K**,**L**) were analyzed. In all graphs, data from animals that received the experimental vector (blue), and the control vector (red) are represented. Data are expressed as mean per mouse ± SEM. Student’s un-paired *t*-test: * *p* < 0.05 and *** *p* < 0.001. Number of C57BL6/6J mice: 12 females and 6 males (pHIV-Fgfr4 vector) and 12 females and 7 males (pHIV-mCherry vector). Analyses were performed 6 weeks after SCI. AUC: Area under the curve. Scale bar (**A**–**F**): 1 mm.

**Figure 5 cells-12-00528-f005:**
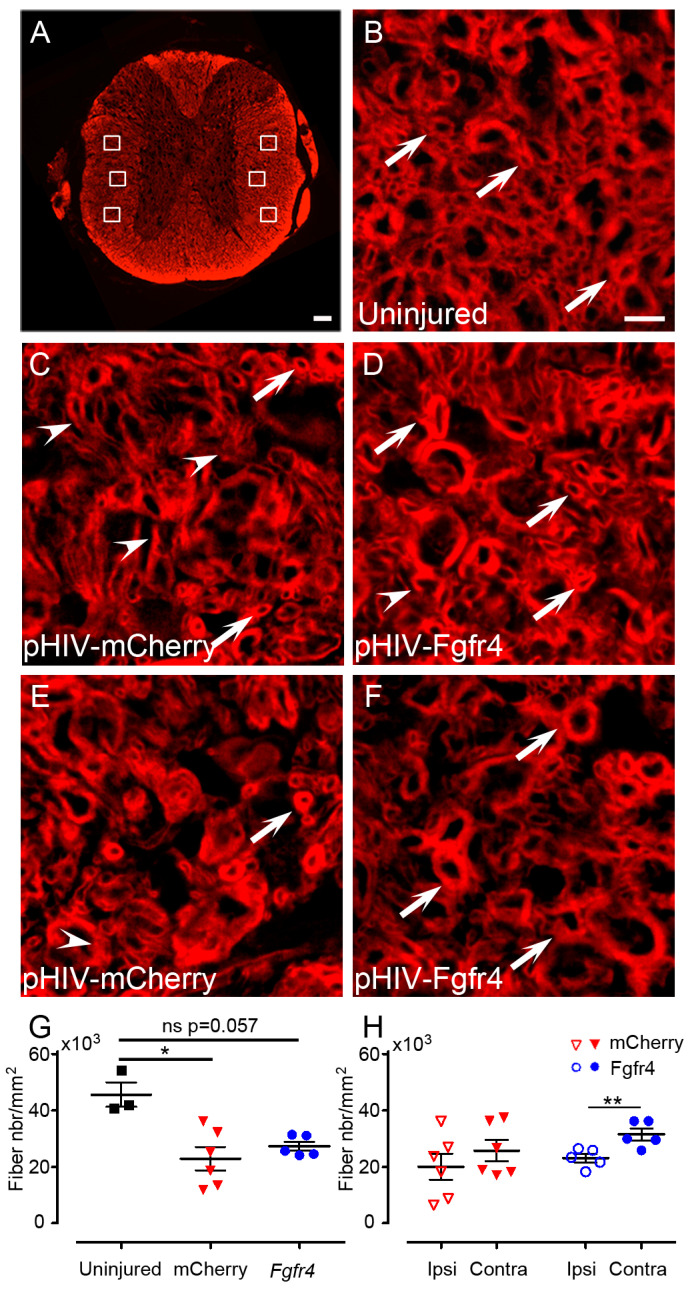
Quantification of spared myelin after SCI and pHIV-Fgfr4 injections. Axial section of a mouse spinal cord stained with fluoromyelin (**A**). White boxes indicate locations of the 6 high magnification images acquired per sections for quantifications of spared myelin fibers (**A**). Representative image of the white matter in an uninjured mouse (**B**). Representative images of the white matter in pHIV-mCherry (**C**,**E**), and pHIV-Fgfr4 (**D**,**F**) animals at 5 mm caudal to the lesion. Representative images on the ipsilateral (**C**,**D**) and contralateral (**E**,**F**) side of the lesion. Arrowheads point to damaged myelin fibers (uncomplete myelin sheath; (**C**)–(**E**)) and arrow point to spared myelin fibers (complete myelin sheath (**B**)–(**F**)). Quantifications were performed on 1 section located 3 mm rostral and 1 section located 3 mm caudal to the lesion epicenter. Density of the overall spared myelinated fibers (rostral and caudal; ipsilateral and contralateral; equivalent age and locations matches were taken for uninjured) (**G**). Density of spared myelinated fibers (rostral and caudal) ipsilateral and contralateral to the lesion site (**H**). Data from uninjured animals (black), animals that received the experimental vector (blue), and the control vector (red) are represented. Data are expressed as mean per mouse ± SEM. Student’s un-paired *t*-test: * *p* ≤ 0.05 (**G**) and paired *t* (**H**) ** *p* ≤ 0.01. Number of female C57BL6/6J mice: 3 uninjured, 6 control (pHIV-mCherry), and 5 experimental (pHIV-Fgfr4). In both groups (pHIV-mCherry and pHIV-Fgfr4) spinal cords were analyzed 6 weeks after SCI and vector injections. Scale bars: 100 µm (**A**) and 50 µm (**B**–**F**).

**Figure 6 cells-12-00528-f006:**
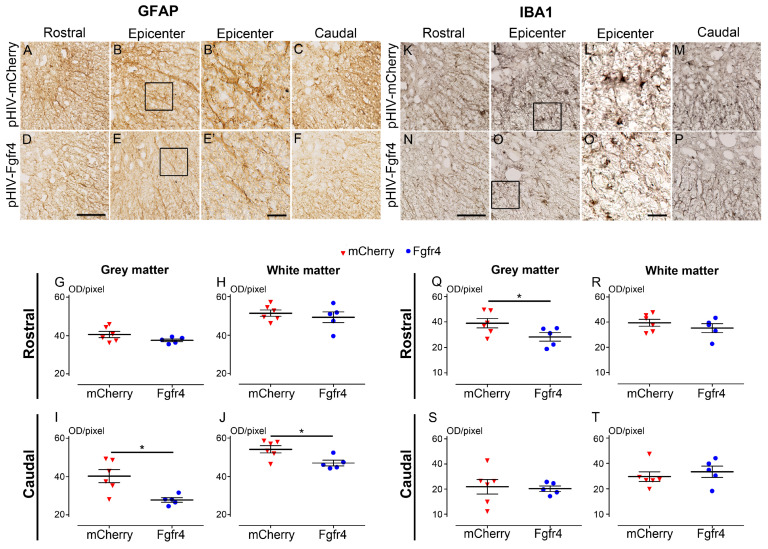
pHIV-Fgfr4 vector injection immediately after SCI reduces astrocyte reactivity caudal to the lesion. Glial reactivity was assessed using peroxidase immunohistochemistry for astrocyte (GFAP; **A**–**F**,**B’**,**E’**), and microglia (IBA1; **K**–**P**,**L’**,**O’**). Brightfield micrographs of GFAP in the injured spinal cords of mice injected with the control vector (**A**–**C**) or the experimental vector (**D**–**F**). Micrographs were taken at the lesion epicenter on the contralateral side (**B**,**E**), rostral (**A**,**D**), and caudal (**C**,**F**) to the lesion site. Zoomed-in micrographs taken in the lesion site of the control (**B’**) and the experimental (**E’**) group (black insets in (**B**) and (**E**), respectively)**.** Quantifications of GFAP expression were performed rostral (**G**,**H**) and caudal (**I**,**J**) to the lesion site in the grey (**G**,**I**) and the white matters (**H**,**J**). Immunoperoxidase staining for IBA1 (**K**–**P**) was performed in the injured spinal cords of mice injected with the control vector (**K**–**M**) or the experimental vector (**N**–**P**). Micrographs were taken at the lesion epicenter on the contralateral side of the lesion (**L**,**O**), rostral (**K**,**N**), and caudal (**M**,**P**) to the lesion site. Zoomed-in micrographs taken in the lesion site of the control (**L’**) and the experimental (**O’**) group (black insets in **L** and **O**, respectively). Quantifications of IBA1 expression were performed rostral (**Q**,**R**), and caudal (**S**,**T**) to the lesion site in the grey (**Q**,**S**), and the white matters (**R**,**T**). In all graphs, data from animals that received the experimental vector (blue), and the control vector (red) are represented. Data are expressed as mean per mouse ± SEM. Student’s unpaired *t* with Welch’s correction * *p* ≤ 0.05. Number of C57BL6/6J female mice: 5 pHIV-Fgfr4 and 6 pHIV-mCherry. Analyses were performed at 6 weeks after SCI. Scale bars: 100 µm (**A**–**F**,**K**–**P**) and 25 µm (**B’**,**E’**–**L’**,**O’**).

**Figure 7 cells-12-00528-f007:**
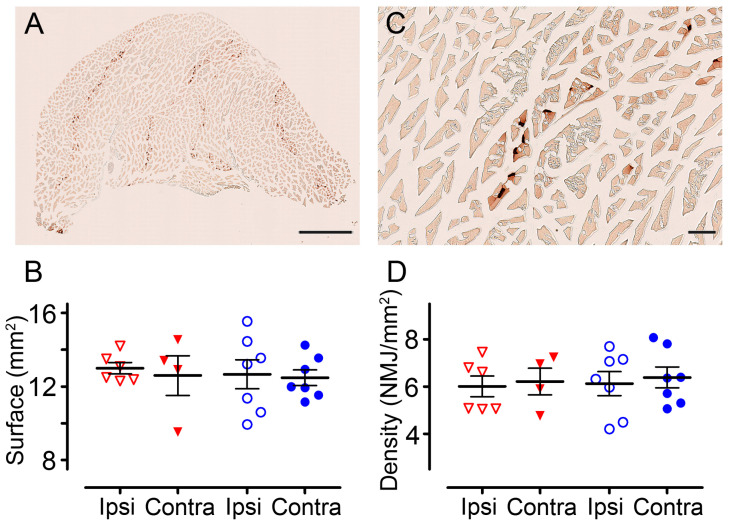
Quantification of neuromuscular junctions in the *gastrocnemius-soleus-plantaris* muscular complex. Bright-field micrographs showing gastrocnemius–soleus–plantaris muscle complex of the hind limb ipsilateral to the spinal cord lesion in an untreated mouse (**A**). Quantification of muscle surface area assessed in pHIV-mCherry (red) and pHIV-Fgfr4 (blue) groups ipsilateral and contralateral to the lesion side (**B**). Neuromuscular junctions (**C**) quantification in muscles of pHIV-mCherry and pHIV-Fgfr4 mice were performed ipsilateral and contralateral to the lesion side (**D**). Data are expressed as mean per mouse ± SEM. Student’s unpaired *t* with Welch’s correction and paired *t*-test (comparison ipsi vs. contra). Number of female mice: 5 C57BL6/6J experimental mice; 7 C57BL6/6J control mice. Animals were sacrificed at 6 weeks after SCI. Scale bar: 1 mm (**A**) and 100 µm (**C**).

## Data Availability

All supporting information is available from the corresponding author on reasonable request. Links to publicly archived datasets generated during the study will be provided later.

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
