# Peer review of "Up-Regulation of Astrocytic Fgfr4 Expression in Adult Mice after Spinal Cord Injury"

_cells, 2023, doi:10.3390/cells12040528_

Round 1

Reviewer 1 Report

This is an interesting paper examining the effect of over-expressing FGFR4 on astrocytes after spinal cord hemisection.  They target FGFR4 to astrocytes using a GFAP promoter by lentivirus transduction, even though they show good mCherry expression, FGFR4 expression is not verified.   Expression of BIII tubulin is identified in mCherry + cells previously of astrocyte origin, however, this is not rigorously done.  Behavioral data indicates some differences in gross locomotor function associated with females more so than males.  The authors also used MRI to identify the extent of lesioned tissue indicating that females had shorter lesion extension and volume.  Further analyses indicated increases in complete myelin, reduced GFAP levels caudal and microglia rostral the epicenter and a potential genetic switch in astrocytes in female mice transduced with FGFR4.  

Concerns:

Figure 2, the data showing BIII tubulin expression in mCherry+ cells is not convincing.  From the image it appears that all GFAP+ cells are BIII tubulin+.  Additionally, BIII tubulin staining in mCherry controls are not shown and the data is not quantified to determine the extent of neuronal conversion.

IN Figure 3, it is surprising that total distance in B or the contralateral print position in F are not statistically significant.  The effect being significant in females and not males could be caused by the number of females tested being twice the males and the male sample size could simply be underpowered.  It is suggested that the number of males be increased to match the female groups to determine the extent of sexual dimorphism.  Other behavioral assays such as grid walkway and the BMS would be helpful in separating out sexual dimorphic differences.  Presently only very minor behavioral differences are identified and suggesting they represent gross and fine motor control is a bit of over interpretation.

Figure 5, It is unclear what is considered incomplete myelin and not convincing from the images.  If it is considered degenerative, how it is distinguished from tissue artifact during cutting of sections, which can disrupt myelin profiles.  The measures of total myelin should also be presented irrespective of whether or not it is considered incomplete for Figure 5 G & H.  The best method of determining degenerative myelin is by cutting plastic sections and electron microscopy.   Again the study appears underpowered since the numbers of FGF

Since the FGFR2 lentivirus was injected into the dorsal columns, why were myelin measures and many histological measures done in the lateral funiculus?  Did they identify mCherry expression within that location.   Lower magnification of the area expressing mCherry would be helpful.  It would also be helpful to see the expression of FGFR4 immunohistochemistry.  

Minor concerns:

Adjust X axis in figure 3 to match the proper notches.

How was mCherry expressed in the FGFR4 virus, fusion? P2A?, IRES?  This could impact expression levels of either transgene or possibly block receptor activation if a fusion construct.

In Figure 6, the microglia (IBA1) staining has a high level of background, and the structures are more reminiscent of astrocytes than microglia.

There is some confusion in quantitative data for whether the number of samples represents section number or animal numbers (e.g., figure 5) .

Author Response

We are grateful to the reviewer’s comments that help to improve the quality of our manuscript. We have addressed all her/his questions/comments. Modifications as compared to the previous version can be followed since they are highlighted in yellow in our revised manuscript. In particular, we have added immunohistochemistry acquisition (Figure 2 and Figure 1 and 3 for the reviewer) as well as new analysis of behavioral data (Figure 3 and Figure 2 for the reviewer).

You will find our responses in the attach document.

Reviewer 2 Report

This paper explores the upregulation of Fgfr4 expression in astrocytes after spinal cord injury in adult mice, but the content of the research is scattered and superficial. This idea needs to be reorganized and more experiments added to support the proof.

Author Response

We have improved the quality of our manuscript. Modifications as compared to the previous version can be followed since they are highlighted in yellow in our revised manuscript. In particular, we have added immunohistochemistry acquisition (Figure 2) as well as new analysis of behavioral data (Figure 3). You will find the description of added data in the attached document.

Round 2

Reviewer 2 Report

This is an in-depth and valuable study.This study found that over 10% of 12resident astrocytes surrounding the lesion spontaneously transdifferentiate towards neuronal phenotype. Then further proofed that this conversion is associated with an increased expression of fibroblast growth factor receptor 4 (Fgfr4), a neural stem cell marker, in astrocytes.The foundings suggest that gene therapy targeting Fgfr4 over-expression in astrocytes after injury is a feasible therapeutic approach to improve recovery following traumatism of the spinal cord. In this paper, argue from multiple aspects, which conclusion has high reliability. But the author still needs to solve some problems.

General comments:

1. This study stress that sex-dependent response to astrocytic modulation should be considered for development of effective translational strategies in other neurological disorders.But less is reflected in the final conclusion.

2. Fgfr4 plays a pivotal role this conversion. But this article discuss increased expression of fibroblast growth factor receptor 4, also should simply verify the performance of low expression.

Other comments:

1. Few grammatical errors in the manuscript. English should be edited.